# A Waterborne, Flexible, and Highly Conductive Silver Ink for Ultra-Rapid Fabrication of Epidermal Electronics

**DOI:** 10.3390/s25072092

**Published:** 2025-03-27

**Authors:** Patrick Rwei, Jia-Wei Shiu, Mehmet Senel, Amirhossein Hajiaghajani, Chengyang Qian, Chin-Wen Chen, Peter Tseng, Michelle Khine

**Affiliations:** 1Department of Electrical Engineering and Computer Science, University of California Irvine, Irvine, CA 92697, USA; prwei@uci.edu (P.R.); ahajiagh@uci.edu (A.H.); tsengpc@uci.edu (P.T.); 2Institute of Organic and Polymeric Materials, National Taipei University of Technology, Taipei 10608, Taiwan; bowenshiu@ntut.edu.tw (J.-W.S.); cwchen@ntut.edu.tw (C.-W.C.); 3Department of Biochemistry, Faculty of Pharmacy, Biruni University, 34010 Istanbul, Türkiye; msenel81@gmail.com; 4Department of Biomedical Engineering, University of California Irvine, Irvine, CA 92697, USA; chengyq1@uci.edu

**Keywords:** soft electronic, water-borne conductive ink, silver ink, on-skin electronics

## Abstract

Epidermal electronics provide a promising solution to key challenges in wearable electronics, such as motion artifacts and low signal-to-noise ratios caused by an imperfect sensor–skin interface. To achieve the optimal performance, skin-worn electronics require high conductivity, flexibility, stability, and biocompatibility. Herein, we present a nontoxic, waterborne conductive ink made of silver and child-safe slime for the fabrication of skin-compatible electronics. The ink formulation includes polyvinyl acetate (PVAc), known as school glue, as a matrix, glyceryl triacetate (GTA) as a plasticizer, sodium tetraborate (Borax) as a crosslinker, and silver (Ag) flakes as the conducting material. Substituting citric acid (CA) for GTA enhances the deformability by more than 100%. With exceptional conductivity (up to 1.17 × 10^4^ S/cm), we demonstrate the ink’s potential in applications such as an epidermal near-field communication (NFC) antenna patch and a wireless ECG system for motion monitoring.

## 1. Introduction

The development of wearable technologies over the past several decades has opened new opportunities for personalized, real-time health monitoring, wellness, and safety. While much progress has been made in the design and fabrication of epidermal-like sensors, with flexibility and softness to improve wearability in challenging circumstances such as for ambulatory settings [1,2,3,4], the weak adhesion and inconsistent interface between the sensors and the skin often render wearable electronics susceptible to motion artifacts, making the received data difficult to analyze [5,6,7,8,9,10]. While notable advances have been made in adhesive epidermal patches designed for conducting electrophysiological measurements [5,6,7,8,9] and emerging commercial biosensors [10], their conformability remains hindered by the mechanically stiff auxiliary battery and wireless unit that resides on the rigid PCB board [11,12]. These elements make the patches less flexible and harder to conform to the skin, affecting the overall comfort. A fully conformable epidermal electronic system would improve comfort, facilitate long-term wear, reduce hospital care costs, and increase patient compliance.

Common fabrication methods for flexible sensors, such as digital light processing (DLP)-based 3D printing, have enabled the development of low-density, high-modulus materials that are suitable for creating intricate structures with a high resolution [13]. Conductive inks are a promising approach to overcoming these limitations by enabling seamless sensor–skin integration [14,15]. Applying conductive inks directly to the skin eliminates gaps or wrinkles between the skin and sensors, ensuring more precise, motion artifact-free sensor data without requiring additional hardware or calibration. However, achieving these advantages necessitates inks that are not only highly conductive and flexible, but also biocompatible and safe for direct skin application.

Recently, research in conductive inks and printing motifs inspired by inkjet, screen, and stencil methods has addressed the need for versatile design options and seamless component interfacing [16,17]. Recent studies have investigated the use of conductive materials, such as PEDOT: PSS and MXene, to improve the performance of flexible sensors [18,19]. PEDOT: PSS, a conductive polymer, is widely used due to its conductivity and flexibility. However, it suffers from low mechanical stability, limited biocompatibility, and less conductivity compare to metals. MXene materials, known for their excellent electrical conductivity, high flexibility, and mechanical properties, have emerged as a promising alternative for wearable electronics. However, they face challenges related to complex synthesis, surface contamination, and also a lack of conductivity compare to metals, which can limit their widespread application in ECG monitoring. Liquid metal-based inks are widely used for printing flexible electronics circuits and electrodes. However, liquid metal devices often develop low-conductivity oxide skins and exhibit liquid-polymer stiffness mismatches, compromising their robustness and biocompatibility [20]. This has shifted focus toward skin-safe conductive ink formulations composed of conductive metal precursors such as silver (Ag), copper (Cu), carbon (C), or gold nanoparticles (AuNPs), stabilized with nontoxic polymer bases and binders [21]. Among these, silver nanoparticles (AgNPs) have been known to be advantageous for their high conductivity. Stretchable silver ink enables the creation of circuits that conform to non-planar surfaces, making them ideal for wearable electronics, biomedical devices, and other dynamic applications. However, most AgNPs-based conductive inks are synthesized using toxic solvents [22,23,24]. Waterborne inks are particularly desirable due to their non-toxicity, low boiling points, and ability to process at lower temperatures, reducing substrate damage and enhancing design flexibility. However, most waterborne conductive inks lack the necessary combination of flexibility and conductivity.

Our group has recently developed a waterborne-based ink with outstanding properties in conductivity, flexibility, and biocompatibility. The ink was applied onto an insulating adhesive as a body epidermal area network for wireless data and power transfers during everyday activities, including underwater activities such as swimming [25]. Despite these advancements, further improvements in conductivity and flexibility are essential to enhance the ink’s performance and extend its application potential.

In this study, we report the synthesis of a waterborne Ag ink with high conductivity and deformability, as demonstrated in Figure 1. Inspired by the child-safe toy slime, derived from school glue (Elmer’s Glue-All), we developed a biocompatible Ag waterborne ink that can be painted directly onto the skin or used to fabricate key electronic components, including sensors, interconnections, wires, and electrodes [12]. The ink could achieve a resistivity of 8.52 × 10^−5^ Ω·cm due to the low-temperature sintering process during fabrication, which is an order of magnitude better than other comparable waterborne inks [26,27,28,29,30,31,32,33,34,35,36]. Compared to existing wearable bioelectronics, the fabrication of this ink is simpler and does not require specialized equipment, such as lithography or inkjet printers [37,38,39,40]. Its customizable designs enable the creation of epidermal sensors for diverse applications. To demonstrate the versatility of this waterborne ink, we present its applications as: (1) an epidermal NFC antenna and (2) ECG electrodes and interconnects. Human studies show that this ink is capable of wireless communication as NFC antennas and as paint-on ECG sensors with accuracy comparable to commercial standards, even when the person wearing it is in motion.

## 2. Materials and Methods

### 2.1. Materials

The matrix used was the water-based polymer, polyvinyl acetate (PVAc), commercially available as Elmer’s Glue-All Multi-Purpose Liquid Glue. Silver flakes (Ag) with a mean particle size of 8–12 μm were obtained from Inframat Advanced Material (Manchester, CT, USA). Glyceryl triacetate (GTA, 99%), sodium tetraborate decahydrate (borax, 100%), and acetic acid were from Alfa Aesar (Stoughton, MA, USA), Fluka™ (Kansas City, MO, USA), and J.T.Baker (Phillipsburg, NJ, USA), respectively. SYLGARD^®^ 184 polydimethylsiloxane (PDMS) was sourced from Merck KGaA (Darmstadt, Germany), and PDMS sheets for the substrate were purchased from AAACME (Phoenix, AZ, USA). Citric acid (CA, 99.7%) was purchased from J.T.Baker. All Ag flakes, solvents, and reagents were of analytical grade, and deionized water (DI water) was used in all experiments.

### 2.2. Preparation of Silver Inks

As shown in Figure 1a, silver inks were prepared by mixing Glue-All, GTA, acetic acid, and Ag flakes using a THINKY MIXER AR-100 (Thinky, Tokyo, Japan) at 1000 rpm for 3 min. For the silver ink formulation, borax (5% in water, corresponding to 1% of PVAc + GTA or CA) was added to the mixture and mixed at 1000 rpm for an additional 3 min (Appendix A). The control groups (without borax) were labeled as follows: Group Glue All (PVAc only, Glue-All), Group GTA (PVAc + GTA at a weight ratio of 90:10), Group GTA + Ag (Group GTA + 40% w/w Ag flakes), Group CA (PVAc + CA at a weight ratio of 80:20), and Group CA + Ag (Group CA + 40% *w*/*w* Ag flakes). For samples containing borax, the following groups were prepared: Borax1, Borax2, and Borax3, which consisted of PVAc + GTA/Borax + Ag flakes at PVAc:GTA ratios of 90:10, 80:20, and 70:30, respectively, and Borax4, Borax5, and Borax6, which consisted of PVAc + CA/Borax + Ag flakes at PVAc:CA ratios of 90:10, 80:20, and 70:30, respectively. The Ag flake content was consistently maintained at 41% ± 2% (*w*/*w*). The detailed weight ratios for each sample are provided in Appendix A.

### 2.3. Printing of Conductive Ink

#### Fabrication of Flexible Conductive Devices

As shown in Figure 1b, the sensor fabrication starts with a PDMS sheet along with a removable protective PET film (thickness 0.25 mm). The designed pattern was then cut through the PET protective film by a CO_2_ laser (Versa laser, Scottsdale, AZ, USA), which served as a shadow mask. After laser cutting, the ink pattern was created through stencil printing by applying the ink on the mask, spreading it evenly with a squeegee, and peeling off the stencil. The device was then placed in an oven and cured at 100 °C for 30 min. Finally, the device was encapsulated by pouring and then curing a thin layer of liquid PDMS over it.

### 2.4. Instrumentations

X-ray diffraction (XRD) patterns were acquired by Malvern Analytical (Worcestershire, UK), Empyrean (X’Pert3 Powder). The surface morphology and the element mapping images were determined via a scanning electron microscope (SEM, Hitachi High-Tech Corporation, Tokyo, Japan) with energy-dispersive X-ray spectroscopy (EDS). The tensile behavior was investigated via a uniaxial tension testing machine (COMETECH, Taipei, Taiwan), during which samples were prepared as rectangle-like slabs of PDMS with a 50 mm length, 10 mm width, and 1 mm thickness. The elongation rate was maintained at 25 mm/min. The test was conducted at room temperature. The electric properties such as resistivity, conductivity, resistance, and sheet resistance were measured by a four-point probe meter (MCP-T600, Loresta-gp, Mitsubishi Chemical, Tokyo, Japan). Strain was applied by a motorized linear actuator (Zaber X-NA08A25-E09, Vancouver, BC, Canada). Resistance changes were continuously recorded using a precision LCR meter (Keysight Technologies E4980AL, Santa Rosa, CA, USA).

### 2.5. ECG Applications

The ECG electrodes were directly painted onto the skin using ink stored in a syringe. Sticker-type conductive wires were fabricated as described in the Section Fabrication of Flexible Conductive Devices.

After obtaining the desired wire pattern on a PDMS substrate, the pattern was encapsulated with a layer of medical-grade polyurethane film (3M™ Tegaderm™, Maplewood, MN, USA). Due to the reduced adhesion between the ink and the PDMS substrate, the ink pattern could be easily peeled off and transferred to the Tegaderm, forming the sticker-type conductive wire. The ECG flexible PCB boards included an ECG front-end chip (AD8232) for signal collection and filtering, and a low-energy Bluetooth chip with an integrated microcontroller (NRF52832, Oslo, Norway) for data transmission.

## 3. Results and Discussion

### 3.1. Characterizations

The effects of sintering at 100 °C for 30 min on the crystallographic profile of the synthesized ink was characterized by XRD (Appendix A). The Ag-flake-added ink, group Borax1 (PVAc + GTA + Ag flake), shows diffraction peaks consistent with the face-centered cubic Ag (JCPDS 04–0783, Ag-flake). No impurity phases were detected [41], indicating that the Ag-flake crystal structure was preserved after sintering. Additionally, the intensity of the Ag diffraction peaks increased after sintering at 100 °C compared to the non-sintered sample (cured at room temperature), suggesting a higher degree of crystallinity or a larger Ag crystal size. Appendix A illustrates the resistance at different curing temperatures, showing that higher curing temperatures result in lower resistance. This behavior may be attributed to a larger conductive path due to increased crystallinity and reduced porosity at elevated temperatures. As expected, the control group (PVAc only, Glue All) exhibited no diffraction peaks, consistent with its amorphous and particle-free nature.

The morphology of the synthesized ink was characterized using SEM and EDS mapping (Figure 2). Compared to pure Elmer’s Glue-All (Appendix A) and its mixture with GTA (Appendix A), the addition of Ag flakes to the Elmer’s Glue-All and GTA matrix significantly increased the surface roughness. Micron-sized features were observed, as shown in Figure 2a, and EDS mapping confirmed that these features correspond to Ag flakes in the GTA + Ag group. These results demonstrate the successful integration of Ag flakes into the Elmer’s Glue-All and GTA matrix. Furthermore, compared to the group cured at 100°C (Borax5, Figure 2b), samples cured at room temperature (Figure 2d) exhibited greater spacing between Ag flakes, indicating higher porosity and correlating with increased resistance. SEM and EDS mapping (Figure 2c) also revealed increased spacing between Ag flakes after stretching.

### 3.2. Mechanical Properties

The mechanical properties of the prepared inks are given in Table 1. Pure Elmer’s Glue-All (group Glue All) exhibited the highest Young’s modulus and lowest fracture strain, indicating high brittleness and low stretchability. The addition of GTA as a plasticizer [42,43,44,45] into the Elmer’s Glue-All (group GTA) decreased Young’s modulus by three orders of magnitude and enhanced its elongation by two orders of magnitude. GTA, a biocompatible and cost-effective plasticizer commonly used in food additives, perfumes, and cellulose [46,47,48], disrupts polymeric interactions, acting like a solvent when mixed with a polymer (Figure 1a). This reduces cohesion between polymer chains, lowering the tensile strength and rigidity [49,50].

This is supported by our results. The further addition of Ag flakes as a conducting material (group GTA + Ag) preserved the enhancement in elongation. To demonstrate that the plasticizer effect can be achieved not only by GTA, citric acid (CA) was also studied as a plasticizer [51]. Substituting the GTA plasticizer with CA (group CA) with the addition of Ag flakes (group CA + Ag) showed similar trends in the decrease in the Young’s modulus and increase in elongation when compared with pure glue (group Glue All). Moreover, CA-embedded ink (group CA + Ag) demonstrated a lower Young’s modulus than GTA-embedded ink (group GTA + Ag), demonstrating higher mechanical flexibility. These results highlight the ability of plasticizers to enhance the flexibility of the base material (i.e., Elmer’s Glue-All), preserved upon the addition of the conductive filler, Ag flakes.

To enhance conductivity, borax was added as a crosslinker [52,53]. Appendix A illustrates the interaction between borax, GTA, and PVAc, where borax crosslinks the PVAc polymer chains. Two types of boron sites are present: tetra-coordinated boron and tri-coordinated boron (Figure 1a) [54,55]. The tetracoordinated borate ions in the solution can react with PVAc and crosslink the PVAc together to form a highly crosslinking gel with rubber-like properties when the pH > 7 [56]. Acetic acid was used to favor the formation of tetra-coordinated boron species and achieve an optimal crosslink density. The mechanical properties of borax-crosslinked inks are shown in Table 1.

Samples Borax1, Borax2, and Borax3 are borax-crosslinked and correspond to PVAc:GTA ratios of 90:10, 80:20, and 70:30, respectively. Sample Borax1 and Sample GTA + Ag differ only in their borax content (PVAc:GTA ratio of 90:10). Sample Borax1 exhibited a higher Young’s modulus and lower elongation when compared with sample GTA + Ag, supporting borax’s role as a crosslinker. Upon the enhancement of the GTA content (Sample Borax2, PVAc:GTA = 80:20), the modulus decreased by 2.67 fold, highlighting GTA’s effect as a plasticizer. The modulus was further lowered with the increased GTA content (Sample Borax3, PVAc:GTA = 70:30), showing increased mechanical flexibility. Based on these results, Figure 1a shows our hypothesized microscopic structure of the ink: the tetracoordinated borate ions crosslink the PVAc polymer chains, and GTA fills between the polymer chains as a plasticizer that enhances mechanical flexibility. The increase in elongation and decrease in the Young’s modulus were observed with CA as the plasticizer (Table 1). Compared with GTA-containing borax samples, CA-containing borax samples showed higher flexibility with PVAc:CA ratios of 80:20 (group Borax5) and 70:30 (group Borax6). These results show that biocompatible waterborne-based Ag-flake ink with high flexibility and suitable conductivity is achieved using CA as a plasticizer. The schematic in Figure 1a illustrates the chemical interactions, where the acidic properties of CA facilitate the formation of tetra-coordinated boron species. Additionally, CA directly interacts with borax, effectively acting as both a plasticizer and crosslinker [57]. In summary, CA enhances mechanical flexibility and ductility by serving as a plasticizer, crosslinker, and acid.

### 3.3. Electrical Properties

The ink exhibited optimal resistivity when cured at 100 °C (Appendix A). Electrical properties, including resistivity, conductivity, resistance, and sheet resistance, were measured using a four-point probe meter after sintering at 100 °C for 30 min, as summarized in Table 1. Electrical properties were not detected in groups Glue All, GTA, and CA because these formulations lacked Ag flakes. The lowest resistivity was achieved with a PVAc:GTA ratio of 90:10 in both GTA-containing inks (group Borax1) and CA-containing inks (group Borax4). Increasing the GTA or CA content slightly increased the resistivity (Table 1). Figure 3a compares the elongation and resistivity for Borax-containing samples (Borax1–Borax6). Among these, sample Borax5 was selected as the targeted ink due to its excellent ductility and good conductivity. All values represent the mean of five samples (*n* = 5).

### 3.4. Properties of the Flexible Conductive Devices

Flexible conductive devices were fabricated by screen-printing water-based Ag ink (PVAc:CA = 80:20) onto a PDMS substrate (Figure 4a,b). Figure 4c depicts the cyclic bending test of the device shown in Figure 4a over 1500 cycles. Strain was applied using a motorized linear actuator at a rate of 20 μm/s until reaching the electrical strain-to-failure point. The film demonstrated excellent tensile elongation, with a break elongation of over 200%, as shown in Figure 4d. Resistance changes were continuously recorded throughout the straining process. Figure 4e,f illustrate the relative resistance of the serpentine device over 1000 cycles under strains between 0% and 20%. Stability was achieved after 100 stretching cycles. This behavior is attributed to the hysteresis of the particle-filled polymer architecture, where the density of Ag flakes decreases in the stretched state [58,59,60,61,62]. Figure 4f provides a zoomed-in view of the cycling test, highlighting the stretching position and demonstrating good responsiveness, with an 81 ms response time and a 93 ms relaxation time. These findings confirm that the water-based conductive Ag ink offers high flexibility without compromising its electrical performance.

## 4. Application

### 4.1. Epidermal NFC Coils Bending Test

To optimize the performance, the coil array was positioned midway between the inner and outer elbow-bending regions to minimize excessive mechanical pressure and stretch. Figure 5 illustrates the bending test of epidermal NFC coils at different angles. The uneven stretching of the flexible coils can cause resonance frequency shifts from the target 13.56 MHz (Figure 5b). To prevent performance degradation, the coils were pre-stretched by 10%. The proposed mechanically flexible ink enables the magneto-inductive coil array to withstand pre-stretching without significantly affecting the NFC performance.

### 4.2. Paint-On ECG Electrode and Ink Wires

The elastic silver ink can be applied not only in epidermal electronics for wearable health monitoring, but also as surface electrodes for biopotential acquisition through direct epidermal contact. A wireless, flexible ECG monitoring patch was fabricated by painting the electrodes directly onto the skin with Ag waterborne ink (Figure 6a). Wire leads were fabricated by stencil-printing the ink onto Tegaderm and encapsulated on both sides for insulation. Appendix A presents the stability of the painted electrodes, while Appendix A depicts the flexible wireless ECG board (AD8232 biopotential integrated circuit). The system was encapsulated with a medical-grade adhesive film, providing a waterproof, sterile barrier against external contaminants. The analog voltage signal obtained from the patch is shown in Figure 6b. ECG signals were monitored wirelessly for 30 min, with no observed skin irritation during the trial (Appendix A). This indicates conformal skin contact, supported by a high signal-to-noise ratio compared to an FDA-cleared physiological monitoring system with wet electrodes (BIOPAC MP36). Both datasets were processed in MATLAB (MATLAB R2024b) using a 4th-order Butterworth low-pass filter with a cutoff frequency of 30 Hz (Figure 6b).

The advantages of paint on ECG in eliminating the motion artifact are shown in Figure 6c. Appendix A shows the setup of commercial ECG wires and pads compared to our paint on wireless ECG application. The subject starts from walking to running with the setups during the measurements. The data collected from the commercial wires and pads were noisy because of the bulky wires, adhesive with the skins, and motion artifact compared to the paint on ECG. Figure 6c shows the filtered ECG data of both setups. From the enlarged extracted running ECG waveform, the characteristic ECG peaks (P, Q, R, S, and T) were distinctly clear and consistent in our paint on setup. In contrast, the bulky wire setup was highly susceptible to motion artifacts during running, evidenced by the deformed ECG waveform. Figure 7a shows the heart rate of the paint on ECG setup during running with an Apple watch, a valuable heart rate commercial device [63], monitoring simultaneously. The subject was tested by walking and running cycles every one minute, as shown in Figure 7c. It shows a compromising heart rate result compared to the commercial device. To evaluate the level of agreement between the paint-on heart rate values and the commercial heart rate device values, Bland–Altman analyses were performed (Figure 7b). According to the analysis, the mean bias and standard deviation (SD) were 0.84 ± 1.96. The analysis indicated that data obtained from both measurement methods had a strong correlation. Above all, our paint-on ECG application could perform accurate heart rate monitoring and clear ECG signals with good motion artifact resistance.

Future research could focus on optimizing the electrode structure to improve both comfort and signal fidelity. This can involve refining the ink formulation to enhance the durability and elasticity, as well as exploring alternative binders to improve adhesion without compromising conductivity. Improving the biocompatibility of the conductive ink is essential for expanding its use in medical applications, including developing hypoallergenic formulations.

## 5. Conclusions

The development of a waterborne, flexible, and highly conductive silver ink represents a significant advancement in wearable electronics. This study demonstrated the ink’s potential in real-world applications such as epidermal NFC coils and wireless ECG monitoring systems. By incorporating CA as both a plasticizer and crosslinker, the ink achieved significant improvements in elongation and flexibility compared to traditional GTA. The tetracoordinated borate ions formed through the interaction of borax and CA created a highly crosslinked polymer matrix, enabling the ink to maintain its robustness while exhibiting exceptional resistivity (8.52 × 10^−5^ Ω·cm) and conductivity (1.17 × 10^4^ S/cm). Extensive mechanical and electrical testing, including cyclic bending, strain-to-failure, and stretching experiments, validated the ink’s durability under dynamic conditions. The primary advantage of this flexible waterborne silver ink is its ability to flex without losing conductivity. This property allows wearable devices to conform to irregular shapes, such as human skin, making it ideal for applications where traditional rigid PCBs or wires would be impractical. Demonstrations of the NFC bending frequency test and the flexible ECG device showcased the ink’s adaptability, with simple paint-on and sticker-type designs, delivering promising signals with minimal motion artifacts. Moreover, its biocompatibility was confirmed through prolonged skin contact without irritation, making it safe and environmentally friendly.

Despite these achievements, further improvements in the ink’s long-term stability and adhesion under extreme conditions, such as high humidity or extended wear, could broaden its applications. Future work could explore integrating this ink with other wearable technologies, such as biosensors for the continuous monitoring of physiological parameters. Overall, the waterborne silver ink addresses critical challenges in conductivity, flexibility, and biocompatibility, paving the way for innovations in healthcare, sports, and personal monitoring devices. Its ability to conform to irregular surfaces while maintaining its electrical performance establishes a foundation for developing more accessible, reliable, and sustainable wearable technologies.

## Figures and Tables

**Figure 1 sensors-25-02092-f001:**
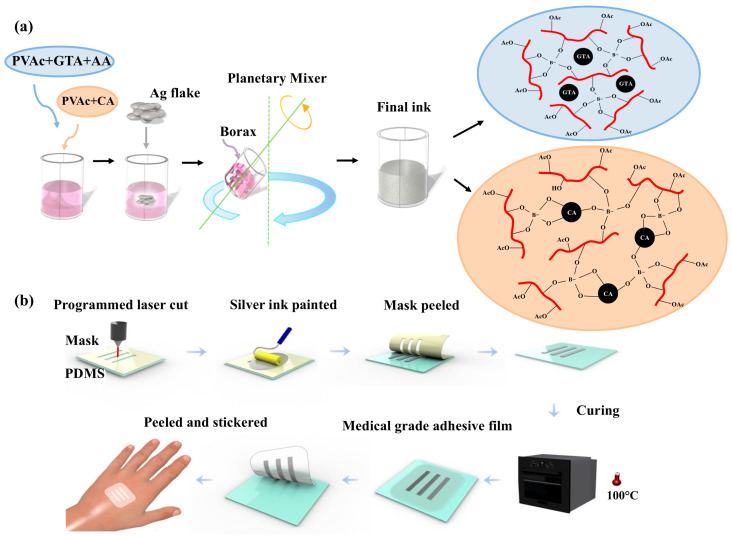
(**a**) Silver ink was prepared by mixing PVAc, GTA, acetic acid, and Ag flake powder at 1000 rpm for 3 min. Borax was then added to the mixture. (**b**) Sensor fabrication began with PDMS sheets covered by a plastic layer. The designed pattern was then cut using a CO_2_ laser. After laser cutting, the ink pattern was created by peeling off the mask and applying the ink via a screen-printing process. The device was then placed in an oven and cured at 100 °C for 30 min. Once cured, the device was coated with a medical-grade adhesive film, allowing it to be peeled off and applied directly to target areas.

**Figure 2 sensors-25-02092-f002:**
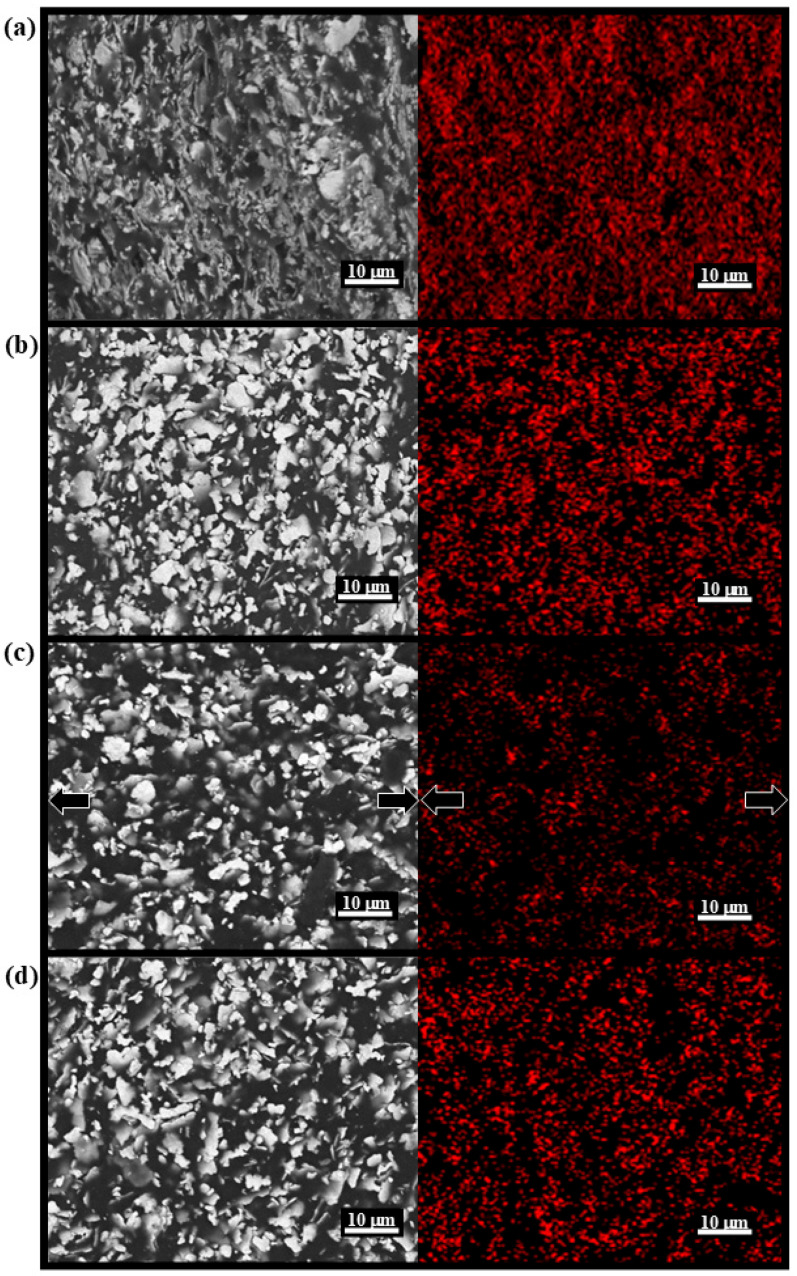
SEM and EDS mapping (Ag) images of (**a**) Group GTA + Ag; (**b**) Group Borax5 with PVAc, CA, Borax, and Ag flakes, cured under 100 °C; (**c**) Group Borax5 during stretched (the arrow indicates the direction of stretch); (**d**) Group Borax5 cured under room temperature.

**Figure 3 sensors-25-02092-f003:**
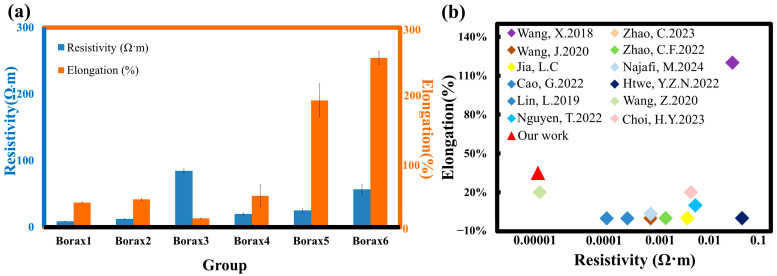
(**a**) The comparison plots for samples with Borax and Borax1-Borax6; (**b**) Comparison of elongation and resistivity of different waterborne-based Ag ink references [12,26,27,28,29,30,31,32,33,34,35,36].

**Figure 4 sensors-25-02092-f004:**
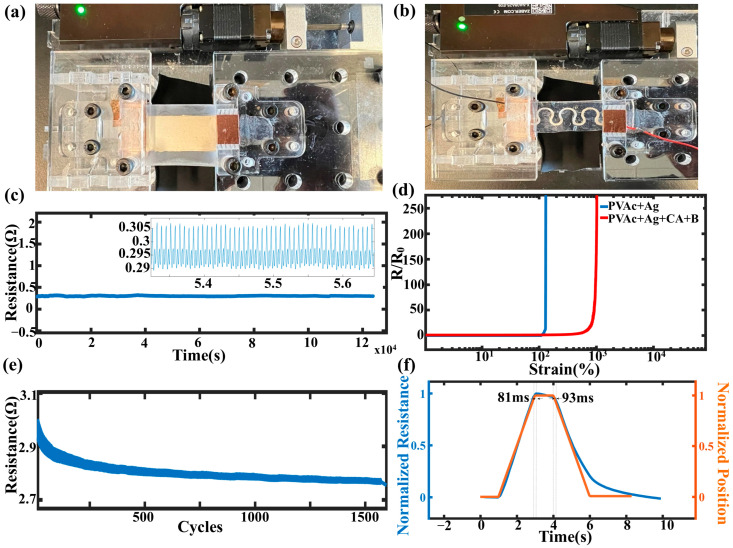
Set-ups for (**a**) flexibility and (**b**) serpentine stretchability tests. (**c**) Cyclic bending test; (**d**) Strain to failure test of the one with and without Borax; (**e**) Relative resistance of the samples under a thousand cycles of strain to failure test; (**f**) Zoomed-in version of the cycling test with the stretching position, showing good responsiveness.

**Figure 5 sensors-25-02092-f005:**
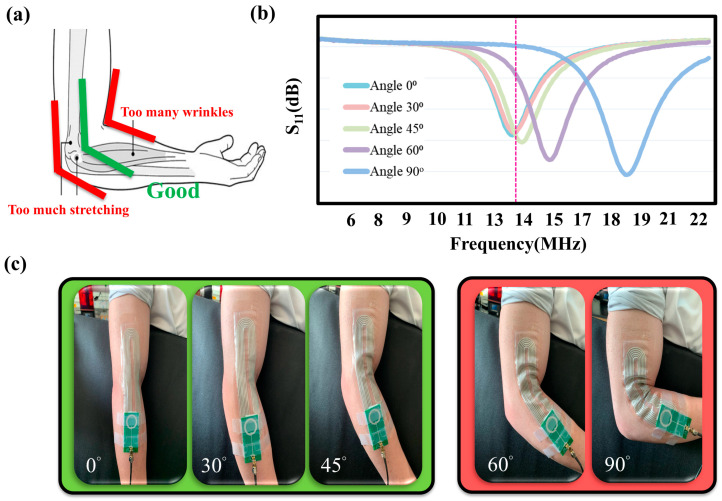
(**a**) The coil array is placed midway in between the elbow inner- and outer-bending regions to avoid excessive mechanical pressure and stretch. (**b**) S11 of the NFC coils during stretch of the dotted line represents the NFC operation frequency of 13.56MHz (**c**) The photos of different stretching angles.

**Figure 6 sensors-25-02092-f006:**
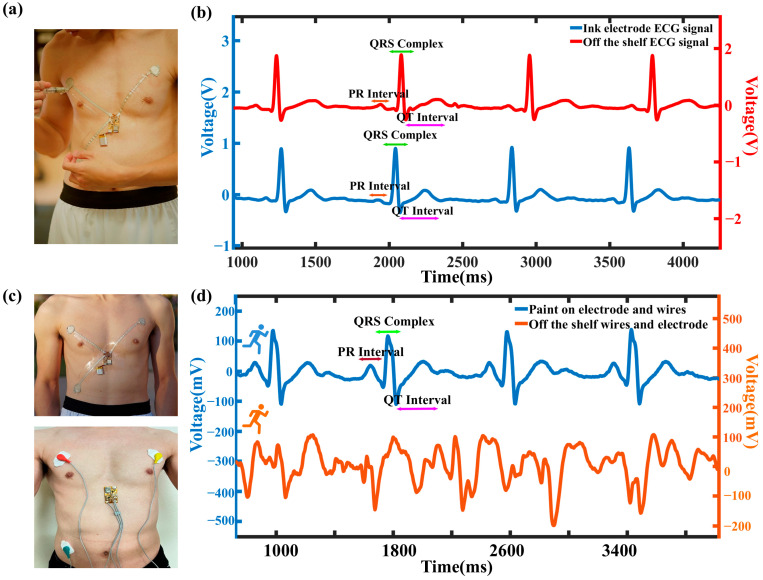
(**a**) The photos of the flexible PCB board, the placement of the silver ink electrode and the board, and the setup of the whole wireless system. (**b**) The filtered ECG signals from the ECG silver ink electrodes and the off-the-shelf ECG device. (**c**) The upper photo shows the setup of the wireless system, and the bottom photo shows the setup of the off-the-shelf wires and electrodes. (**d**) The comparison of the signals from the wireless ECG system and the off-the-shelf wires and pads during running.

**Figure 7 sensors-25-02092-f007:**
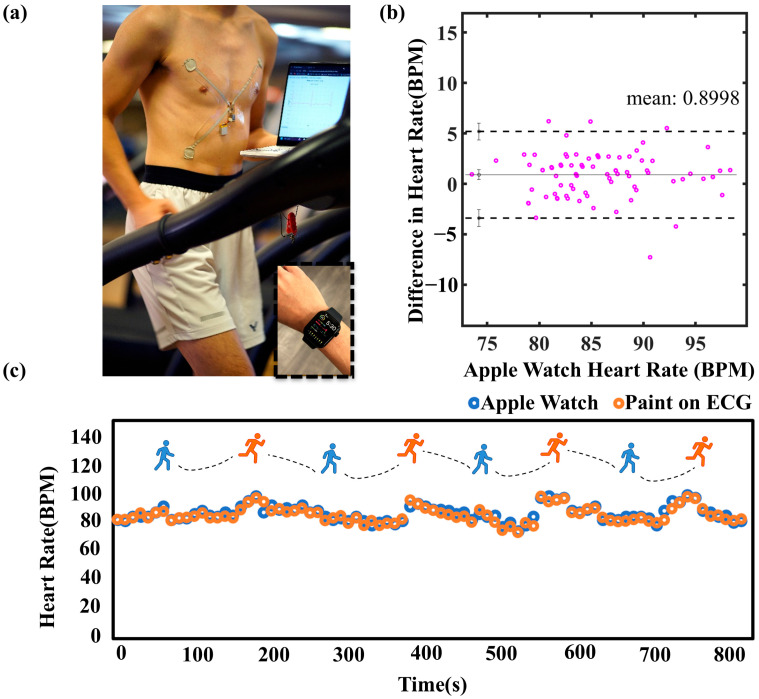
(**a**) Paint-on wireless ECG setups with Apple watch during running. (**b**) Blant–Altman plot of the paint on ECG and Apple watch signals. (**c**) Heart rate obtained from the paint-on ECG and Apple watch during motions.

**Table 1 sensors-25-02092-t001:** The parameters of stress, elongation, resistivity, conductivity, and elasticity coefficient in each sample.

Group	Stress (MPa)	Elongation (%)	Resistivity (10^−5^ Ω·cm)	Conductivity(10^4^ s/cm)	Young’s Modulus (MPa)
Glue All	11.46 ± 2.15	2.88 ± 1.10	--	--	1055.07 ± 517.92
GTA	1.77 ± 0.18	129.9	--	--	5.20 ± 0.43
GTA + Ag	1.85 ± 0.03	75.35	45.82 ± 4.70	0.22 ± 0.02	8.66 ± 0.40
CA	1.78 ± 0.09	258.36 ± 22.49	--	--	0.95 ± 0.21
CA + Ag	1.81 ± 0.15	187.67 ± 15.43	650.20 ± 42.85	0.02 ± 0.01	3.79 ± 0.15
Borax1	2.52 ± 0.05	36.89 ± 1.08	8.52 ± 0.12	1.17 ± 0.02	167.62 ± 22.70
Borax2	1.79 ± 0.04	41.62 ± 1.39	12.08 ± 0.45	0.83 ± 0.03	68.77 ± 22.81
Borax3	0.68 ± 0.05	13.13 ± 0.17	84.34 ± 3.06	0.12 ± 0.01	18.81 ± 4.25
Borax4	8.12 ± 1.19	46.92 ± 16.53	19.59 ± 1.77	0.54 ± 0.04	832.16 ± 51.05
Borax5	2.60 ± 0.10	190.56 ± 24.94	24.88 ± 2.57	0.41 ± 0.04	3.64 ± 0.37
Borax6	1.18 ± 0.68	254.33 ± 9.70	56.59 ± 7.18	4.33 ± 7.17	0.91 ± 0.55

## Data Availability

Data are contained within the article and Appendix A.

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
