# Peer review of "A Waterborne, Flexible, and Highly Conductive Silver Ink for Ultra-Rapid Fabrication of Epidermal Electronics"

_sensors, 2025, doi:10.3390/s25072092_

Round 1
Reviewer 1 Report
Comments and Suggestions for Authors
- In line 127, "0.25mm" should be corrected to "0.25 mm" with a space between the number and the unit; please check the entire text for similar issues.
- In line 128, "CO2 laser" should have the "2" in subscript.
- In line 159, "100 oC" is incorrectly formatted. Please correct it.
- Why did the authors choose a 40% W/W ratio of Ag flakes? Did they consider the effect of different Ag flakes contents on the sensing performance?
- In Figure 4, the GF values under different strain conditions need to be further calculated.
- Can the sensing performance of the prepared sensor material be supplemented?
- Figure 7c needs to be redrawn.
- In the Introduction section, more common fabrication methods for flexible sensors should be added. The following reference can be included to improve the related content: Additive Manufacturing, 2025: 104698.
The English can be improved.
Author Response
Please see attached the file.

Reviewer 2 Report
Comments and Suggestions for Authors
The manuscript demonstrates significant innovation in the field of wearable sensors mainly by introducing a new ink formulation (polyvinyl acetate (PVAc), glyceryl triacetate (GTA), sodium tetraborate (Borax) and silver (Ag) flakes). Reading this manuscript, I have identified a few problems:
- References should be formatted according to the author's guide (ACS style).
- Ref 37–38 at line 81-83 are not related to the paragraph description.
- Reference 41 line 162 is not relevant to the paragraph description.
- I did not find references 64 to 80 in the manuscript.
- Few minor typographical errors: e.g. Line 104 and 128 “CO2”; line 159 “100 oC”, etc.
- Figure 1 should be better put at paragraph 2.3.
Author Response
Please see attached the file.

Reviewer 3 Report
Comments and Suggestions for Authors
The authors have demonstrated a waterborne, flexible and highly conductive silver ink for ultra-rapid fabrication of epidermal electronics. The results suggest the potential of this composite for the utilization of ECG electrodes. However, the following comments should be addressed for further clarification:
- Could the authors provide the FTIR spectra of the Silver ink was prepared by mixing PVAc, GTA, acetic acid, and Ag flake powder. It is important to display typical characteristic stretching bands of the hydroxyl group in these hybrid materials.
- ECG data obtained using flexible and highly conductive silver ink should be compared with traditional Ag/AgCl electrodes to assess performance.
- In the introduction section when discussing existing research, it is recommended to delve deeper into the advantages and disadvantages of dry electrode materials, such as PEDOT:PSS and MXene, as well as their practical applications in ECG monitoring. This will help readers better understand the innovation and necessity of this study.
(e.g.,
ACS Applied Materials & Interfaces 2022 14 (34), 39159-39171
DOI: 10.1021/acsami.2c11921
ACS Applied Materials & Interfaces 2017 9 (43), 37524-37528
DOI: 10.1021/acsami.7b09954
Nature Communications volume 15, Article number: 5974 (2024)
https://doi.org/10.1016/j.jece.2025.115716, https://doi.org/10.1016/j.susmat.2025.e01261
)
- In the discussion section, it is recommended that the authors fully discuss the significance and limitations of the experimental results. For example, you can discuss the potential application of conductive silver ink based dry electrode in ECG monitoring, and indicate the possible future research directions of this study (e. g. optimizing electrode structure, improving biocompatibility, etc.).
Author Response
Please see attached the file.

Reviewer 4 Report
Comments and Suggestions for Authors
This paper presents a waterborne, flexible, and highly conductive silver ink for the ultra-rapid fabrication of epidermal electronics. My conclusion is that this is an interesting and valuable work, and it can be accepted for publication after the following mandatory revisions:
- To better highlight the novelty of this work, the authors should include a table comparing their results with previously published literature on conductive silver inks. The table should summarize key aspects such as the fabrication process, materials used, resistivity/conductivity, etc.
- The authors should also include a table comparing their work with commercial inks to provide a broader context for the advancements presented.
- The authors should specify the laser cutting parameters used for patterning the PET, as this information is essential for reproducibility.
- Clarification is needed regarding the dimensions of the samples used for the elongation test. The authors should state whether the sample dimensions were based on any industry-standard ASTM tests and provide justification for their selection and test setup.
Author Response
Please see attached the file.

Round 2
Reviewer 3 Report
Comments and Suggestions for Authors
This manuscript can be accepted for publication
Reviewer 4 Report
Comments and Suggestions for Authors
The authors have satisfactorily addressed all my comments.